# Objectives and outcomes of patient-driven innovations published in peer-reviewed journals: a qualitative analysis of publications included in a scoping review

Marie Dahlberg ![ORCID],[1] Madelen Lek,[1] Moa Malmqvist Castillo,[1] Ami Bylund,[1] Henna Hasson ![ORCID],[1,2] Sara Riggare,[3] Maria Reinius ![ORCID],[1] Carolina Wannheden ![ORCID][1]

¹Department of Learning, Informatics, Management and Ethics, Karolinska Institute, Stockholm, Sweden
²Unit for Implementation and Evaluation, Center for Epidemiology and Community Medicine, Region Stockholm, Stockholms Lans Landsting, Stockholm, Sweden
³Participatory eHealth and Health Data, Department of Women's and Children's Health, Uppsala Universitet, Uppsala, Sweden

**Correspondence to**
Marie Dahlberg;
marie.dahlberg@ki.se

## ABSTRACT

**Objectives** The aim of this study was to gain a deeper understanding of the objectives and outcomes of patient-driven innovations that have been published in the scientific literature, focusing on (A) the unmet needs that patient-driven innovations address and (B) the outcomes for patients and healthcare that have been reported.

**Methods** We performed an inductive qualitative content analysis of scientific publications that were included in a scoping review of patient-driven innovations, previously published by our research group. The review was limited to English language publications in peer-reviewed journals, published in the years 2008–2020.

**Results** In total, 83 publications covering 21 patient-driven innovations were included in the analysis. Most of the innovations were developed for use on an individual or community level without healthcare involvement. We created three categories of unmet needs that were addressed by these innovations: access to self-care support tools, open sharing of information and knowledge, and patient agency in self-care and healthcare decisions. Eighteen (22%) publications reported outcomes of patient-driven innovations. We created two categories of outcomes: impact on self-care, and impact on peer interaction and healthcare collaboration.

**Conclusions** The patient-driven innovations illustrated a diversity of innovative approaches to facilitate patients' and informal caregivers' daily lives, interactions with peers and collaborations with healthcare. As our findings indicate, patients and informal caregivers are central stakeholders in driving healthcare development and research forward to meet the needs that matter to patients and informal caregivers. However, only few studies reported on outcomes of patient-driven innovations. To support wider implementation, more evaluation studies are needed, as well as research into regulatory approval processes, dissemination and governance of patient-driven innovations.

## BACKGROUND

Multiple policy documents emphasise the importance of involving patients in care decisions and processes.[1–3] Also, many patients and their informal caregivers (eg, parents and family members), increasingly empowered by the improved access to information and technology, expect to actively participate in the

---

### STRENGTHS AND LIMITATIONS OF THIS STUDY

⇒ This study addresses a novel and emerging field of research on patient-driven innovations and builds on a comprehensive scoping review of peer-reviewed literature.

⇒ To our knowledge, this is the first qualitative analysis of published literature on patient-driven innovations, exploring the objectives and outcomes of patient-driven innovations.

⇒ There is a risk of under-representation of research on patient-driven innovations as the drivers of innovations are not always identifiable in scientific publications.

⇒ Given that few publications reported on evaluated outcomes, these findings should be interpreted with some caution.

---

decision-making regarding their care.[3] Active participation may be of particular importance for persons living with chronic conditions as they spend a significant amount of time and energy on managing their illness through self-care. Self-care has been defined as 'the ability of individuals, families and communities to promote health, prevent disease, maintain health, and cope with illness and disability with or without the support of a healthcare provider'.[4] Self-care practices involve self-care maintenance (ie, maintaining emotional and physical stability), self-care monitoring (ie, monitoring changes of symptoms and signs) and self-care management (ie, managing the symptoms and signs as they occur).[5] In addition to practising self-care in their daily lives, persons living with chronic conditions are increasingly also engaged in healthcare improvement, where they contribute with their self-care knowledge and experience to improve healthcare processes and services.[6 7] Thus, patients and their informal caregivers are active partners and not only passive recipients of care.

Patients and informal caregivers have developed many innovative solutions to support them in their self-care. For example, an interview study including 500 patients with rare diseases showed that more than half reported developing a solution to improve self-care management of their own diseases.[8] The Patient Innovation website, which was created to gather and share innovations developed by patients and informal caregivers, has listed over one thousand solutions.[9] These numbers give an indication of significant activity by patients and their informal caregivers driving health innovations, often independently of the health system. Health innovations have been defined as 'new or improved health policies, systems, products and technologies, and services and delivery methods that improve people's health and well-being'.[10] In this paper, we use the term patient-driven innovation to describe health innovations that have been initiated and further developed by patients or informal caregivers.[11]

A scoping review performed by members in our research team explored the nature and extent of patient-driven innovations published in peer-reviewed journals in the time period from 2008 to 2020; it showed a clearly increasing publication trend of mainly technological innovations in the 2010s; all innovations concerned the management of chronic conditions.[11] As this is a relatively new field of research, the knowledge base about patient-driven innovations reported in peer-reviewed journals is still limited. Therefore, the aim of this study was to gain a deeper understanding of the objectives and outcomes of patient-driven innovations that have been published in peer-reviewed journals, focusing on (A) the unmet needs that patient-driven innovations address and (B) the outcomes for patients and healthcare that have been reported.

## METHODS
We reanalysed the 96 publications that were included in our previously published scoping review by performing an inductive qualitative content analysis[12] of data concerning the objectives and outcomes of patient-driven innovations.

### Data collection and analysis
The scoping review was performed by following the framework proposed by Arksey and O'Malley,[13] which has been described in our previous publication.[11] We inductively analysed the data by performing a manifest content analysis, which is a suitable method to describe the characteristics of the content by systematically coding the data and staying close to the text, keeping a low level of abstraction throughout the analysis process.[12 14] Three coauthors (MD, ML and MM), with supervision from MR, collaborated in qualitatively analysing the publications. With backgrounds in social and political sciences, bioentrepreneurship and nursing, they brought several perspectives to the analysis.[15] We defined unmet needs as the problems or challenges that the patient-driven

innovations addressed; if the unmet needs were not explicitly described, we interpreted these based on the descriptions of the patient-driven innovations and their purpose. We defined outcomes as any reports of impact of using the patient-driven innovations, including users' self-reported experiences and perceptions. Whereas all publications were analysed for unmet needs, only publications that were original research studies and were based on empirical data, irrespective of study design, were analysed for outcomes. First, the researchers collectively read all publications and extracted text corresponding to the content areas of interest (ie, unmet needs and outcomes) into an Excel spreadsheet; each publication was individually analysed by two researchers (MD analysed all publications; ML and MM each analysed about half of the publications). The researchers then compared and consolidated their data extractions, whereby uncertainties and conflicts were resolved in discussion. The extracted data constituted the unit of analysis. The analyses were performed collaboratively by MD, ML and MM and who created condensed meaning units and labelled these with descriptive codes. The codes were transferred to Miro, an online visual collaboration platform.[16] Thereafter, MD, ML and MM conducted an iterative process of grouping the codes into categories based on similarities in content. The categorisation was discussed among all coauthors and refined until we reached agreement about the level of abstraction and the labelling of the categories. We report the qualitative findings in combination with descriptive statistics specifying the number and percentage of publications that contributed to each of the inductively created categories; descriptive statistics are used to provide a more thorough description of our analysis.[17] Data about the innovations (eg, name, description, study design) that were included in the initial review were also used in the analysis to contextualise the findings.

### Patient and public involvement
This study was performed within the 'Patients in the driver's seat!' research programme, where patient innovators and researchers collaborate as partners in research. One of the coauthors is a patient innovator (SR) who contributed to all stages of the research process.

### Findings
Of the 96 publications included in the scoping review,[11] 13 publications (14%) contained no information that was relevant to the objectives of this study and were therefore excluded. Thus, our analysis included 83 publications, reporting on 21 patient-driven innovations.[18–100] The publications that were excluded concerned the two innovations (*DIY-APS*[101–107] and *PatientsLikeMe*[108–113]) that accounted for most of the publications included (n=49, 59%). Therefore, the exclusion of publications did not lead to the exclusion of any patient-driven innovations from the analysis. The characteristics of all innovations, grouped by the medical conditions that they addressed, are described in online supplemental appendix 1. The

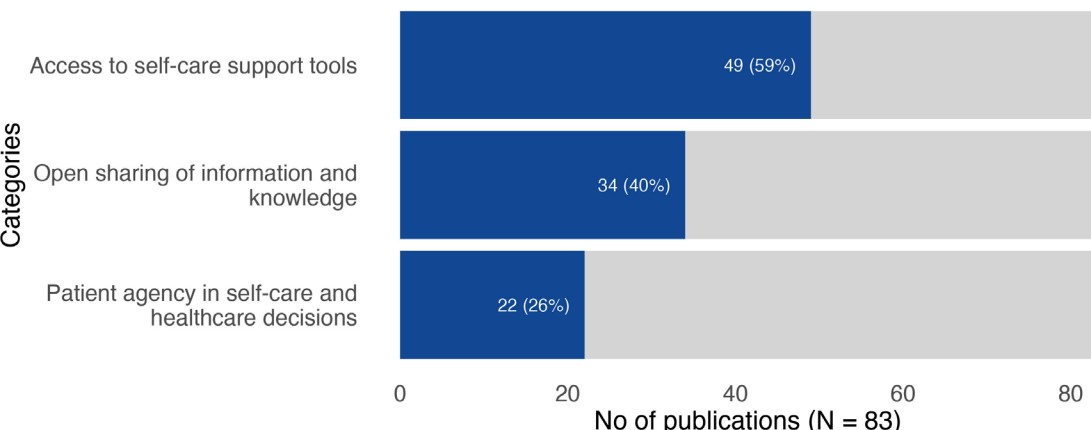

**Figure 1** Number and proportion of publications represented in the three inductively created categories representing unmet needs addressed by patient-driven innovations. NB: some publications were included in more than one category.

coding of each publication is shown in online supplemental appendix 2.

### Unmet needs addressed by patient-driven innovations

All the included publications provided information about unmet needs that were drivers for developing an innovation. We created three categories: access to self-care support tools; open sharing of information and knowledge; and patient agency in self-care and healthcare decisions. Figure 1 illustrates the number of publications represented in each of the categories.

### Access to self-care support tools

A central unmet need identified in over half the publications was access to self-care support for physical as well as social activities. The tools and services that patient innovators developed often supported self-care monitoring. For example, a patient innovator used existing technologies to develop the *Ostom-i-Alert* stoma bag sensor that made it easier to monitor stoma output and prevent leakage.[67–69] To support continuous treatment follow-up, several innovations comprised mobile applications that made self-monitoring data from various medical devices more easily accessible to patients or their informal caregivers.[33 53 54 59 60] In addition to self-care monitoring support, the patient-driven innovations comprised a variety of non-technological products to support physical functioning and participation in social activities. Examples of such self-care management innovations are the *Upsee* harness that supports gait training for children with cerebral palsy,[21 22] and the *Jacki Jacket* that helps patients with breast cancer to hide drainage tubes.[19] We also found how patient innovators came up with new creative self-care methods that involved no or little product development. For example, the *Auditory stimulation* innovation was an auditory interventional innovation to enhance motor function in children with cerebral palsy; it only required access to a musical sound scheme and a device to play music.[20] Similarly, the *3D cue illusions* innovation involved painting a three-dimensional illustration of stairs

on the floor to support gait training for people living with Parkinson's disease.[96]

### Open sharing of information and knowledge

The open sharing of information and knowledge related to self-care was an unmet need addressed in more than one third of the publications. Although most publications concerned the *PatientsLikeMe* platform that enables persons with various health conditions to create personal health profiles, the publications also included innovations with diagnosis-specific platforms (eg, *DIY-APS, Nightscout* and *Webdia* for patients with diabetes type 1). Through these platforms, patients could share their personal health data such as symptoms, treatments and outcomes within the family,[65] or within a wider community of peers.[30 76 78] The platforms thus enabled comparisons of personal health observations with the observations of peers with similar conditions.[80]

In addition to platforms for sharing personal health information, patient innovators developed platforms for open sharing of self-care-related knowledge. For example, the #WeAreNotWaiting-community hosted a platform for sharing do-it-yourself innovations among patients with diabetes type 1; patient innovators could openly share code and provide technological assistance to other patients who wanted to build do-it-yourself solutions.[28 55] In this way, do-it-yourself technologies (eg, *DIY-APS*) could be made available to the broader patient population without requiring regulatory approval.[36 39] Further, this enabled patients to collaborate, contributing to more rapid development and access of innovations.

### Patient agency in self-care and healthcare decisions

An unmet need that was addressed by various innovations, covering several different diagnoses, was patient agency in self-care and healthcare decisions. Patient agency is defined as the requirement of 'skills across the spectrum of participation in care, ranging from active participation in medical encounters and decision-making to self-care skills for managing everyday health-related

activities'.[114] One way to enhance patient agency in self-care was through the curation of trustworthy information and knowledge. Innovations included shared libraries of quality-assured diagnosis-specific self-care information (*T1resources.uk*), as well as personal health records (eg, *MediStori),* and online health information platforms (eg, *PatientsLikeMe*) that provided functionalities for collecting, presenting, and comparing personal health data over time, and with others.[64 72 83 99] These platforms contributed to patient agency in self-care by providing patients with tools to initiate self-monitoring and engaging in self-experimentation, possibly inspired by experiences of other platform users. In addition to tracking personal health data, the *Hallucination tracking* innovation used sensors to collect data from the environment (eg, air temperature, humidity, air pressure); this enabled hypothesis testing to explore correlations between the occurrence of auditory hallucinations and air quality.[99]

Other innovations aimed at improving collaboration with healthcare. For example, mobile patient support systems (eg, *Genia*) have been developed to support partnership between patients, families and their care teams by enabling these actors to directly communicate and exchange relevant information[70 71]; in some cases, also enabling integration of patient-reported data into the patients' electronic medical record.[69 71] A way to increase patient agency at higher levels in strategic healthcare decisions was through the establishment of collaborative multiprofessional networks, involving patients and family caregivers in the design of care pathways,[18] or in the follow-up of therapies.[23 24]

### Reported outcomes of patient-driven innovations
Eighteen (22%) publications were original research studies that were based on empirical data and reported outcomes of the patient-driven innovations. Of these, 5 studies (28%) used an experimental study design, 3 (17%) used an observational study design and 10 (56%) used a descriptive study design. We created two categories representing outcomes: impact on self-care; and impact on peer interaction and healthcare collaboration. Figure 2 illustrates the number of publications represented in each of the categories.

### Impact on self-care
Eleven publications reported impacts on self-care, such as improvements in self-care processes, health outcomes and well-being. Several of the innovations for patients with diabetes type 1 were evaluated with positive outcomes: the *OmniPod* insulin pump was well received by young patients and led to improvements in patients' Hemoglobin A1c (HbA1c) levels[63]; *Webdia* also led to a reduction in HbA1c among a subgroup of young patients struggling with their diabetes control but did not affect health-related quality of life compared with usual care[65]; and patients and informal caregivers using *DIY-APS* reported an overall reduced burden of illness due to less interference of diabetes with daily life activities.[46 50 61] Although healthcare professionals expressed some uncertainty and concerns regarding such do-it-yourself technologies to their patients, they were open to participating in educational training.[42]

Positive impacts on self-care were also reported for innovations that addressed other conditions than diabetes, for example, studies indicated that *auditory stimulation* as well as the *Upsee* harness for gait training led to improved functional motor skills among children with cerebral palsy[20–22]; patients with breast cancer wearing the Jacki Jacket experienced a better body image[19]; and engaging in online social interactions using *PatientsLikeMe* was associated with mental recovery among patients with mental disorders.[73] None of the publications reported any negative outcomes.

### Impact on peer interaction and healthcare collaboration
Seven studies reported impacts on peer interaction and healthcare collaboration. Five of these studies evaluated patients' use and experiences of the *PatientsLikeMe* platform, which contributed mainly to peer interactions. Patients used the platform to ask each other questions, provide advice and form new relationships.[76 77 79] Although using the platform was associated with increased comfort in sharing personal health information with others,[80] one study highlighted potential challenges in online patient communication due to variation in the terminology that patients use for describing symptoms.[81] These findings illustrate how the unmet need for open

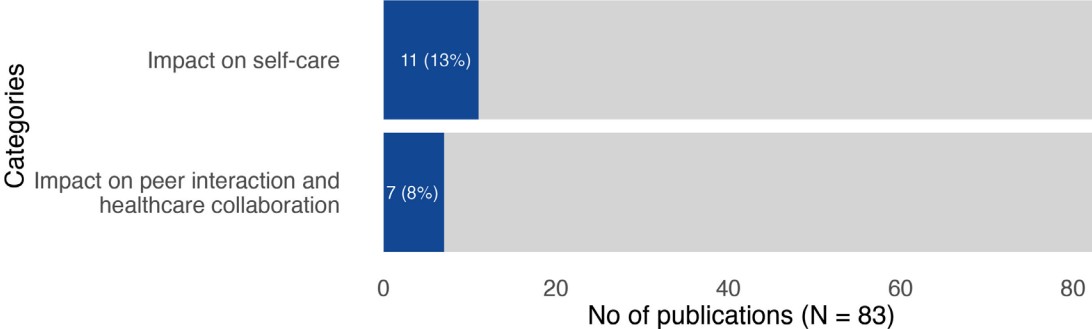

**Figure 2** Number and proportion of publications in the two inductively created categories representing outcomes of patient-driven innovations. NB: in total 65 (78%) of the publications did not report any outcomes.

sharing of information and knowledge among peers was satisfied. Two studies illustrated how patient-driven innovations could also impact collaboration with healthcare, thus strengthening patient agency. The analysis of tweets tagged with #WeAreNotWaiting or #OpenAPS showed that patients as well as healthcare professionals engaged in online conversations about patient-driven diabetes innovations.[30] Active and potential users of the patient support system *Genia* experienced or expected that this patient-driven innovation had the potential to improve communication between patients and their healthcare teams through the systematic collection and reporting of patient-reported observations.[70]

## DISCUSSION
### Principal findings
This study investigated (1) the unmet needs that patient-driven innovations published in peer-reviewed journals have addressed and (2) the outcomes that have been reported. We find that the most frequently identified unmet need was availability and access to self-care support, as indicated by the development and spread of self-care innovations aimed at patients and informal caregivers. Reported outcomes suggested that patient-driven innovations could contribute to improved self-care processes, health outcomes and well-being. Other central unmet needs that patient-driven innovations addressed were the ability to learn from others through information and knowledge sharing, and patient agency in self-care and healthcare. Although fewer studies reported outcomes related to these needs, we identified some studies that indicated positive impacts. None of the studies reported any negative outcomes.

### Strengths and limitations
A strength of this study is its contributions to a novel and emerging field of research on patient-driven innovations by building on a comprehensive scoping review of peer-reviewed literature.[11] Our categorisation of unmet needs addressed by patient-driven innovations and reported outcomes may provide a valuable starting point for future research. Using the scoping review as a basis for the analysis offered a broad inclusion of articles in the field and ensured a systematic search strategy. However, important limitations must be acknowledged. The 21 innovations identified in the scoping review are a mere fraction of the patient-driven innovations that have been openly shared through the Patient Innovation platform, which was developed to lower the 'cost of sharing'.[115 116] It is likely that the relatively few innovations published in peer-reviewed journals are not a representative sample of all patient-driven innovations that are openly available. The dominance of a few patient-driven innovations (eg, *PatientsLikeMe* and *DIY-APS*) that accounted for the majority of publications in our sample can probably be explained by a combination of personal characteristics of the innovators, disease characteristics and opportunities in the respective field.

Thus, we emphasise that the current findings highlight the unmet needs and outcomes only of innovations that are reported in peer-reviewed journals and could be identified as patient driven.

The concept of information power,[117] which guides the assessment of sample size in qualitative studies, is also worth considering when interpreting the trustworthiness of our findings. Information power is affected by several factors, for example, a larger sample size may be needed if the aim is broad and the specificity of the sample in relation to the research question is low. The richness of information that was available in the included publications about the unmet needs addressed by the patient-driven innovations and reported outcomes varied. By complementing our data collection with other data sources, such as personal interviews with the innovators, we would probably have been able to gain more information about the objectives and outcomes of the studied innovations and thereby enhance the information power of our sample. Thus, our findings should be interpreted as indicative, giving rise to hypotheses that merit further exploration. In particular, the reported outcomes of patient-driven innovations should be interpreted with caution as they are based on studies with varying designs (experimental, observational and descriptive). As no negative outcomes were reported, publication bias cannot be excluded and may even be likely. As common for scoping reviews that address broader topics compared with systematic reviews that may include a quite narrow range of quality assessed studies, we did not assess the methodological quality of included studies.[13] Given these limitations, assessing the transferability of our findings to the large number of openly available patient-driven innovations is subject to future research.

### Comparison to prior work
Our findings show that patient-driven innovations addressed a variety of unmet needs related to self-care and healthcare, reinforcing earlier studies that highlight the potential of patients and informal caregivers as important drivers of innovation in healthcare.[8 118 119] By diffusing patient-driven innovations through different channels, including peer-reviewed journals, they have the potential to contribute to unmet needs at a community level. For example, the #WeAreNotWaiting and #OpenAPS movements in diabetes care illustrate how the development and diffusion of patient-driven innovations could radically change the lives of patients, ultimately improving patients' glucose control and reducing caregiver burden.[30 46] The fact that about one third (31%) of the publications included in our analysis concerned the DIY-APS innovation demonstrates how one innovation gained wide attention in the scientific community. This example reflects a culture of open peer-to-peer sharing, speeding up access to innovative solutions.[118] However, it should be considered that only a fraction of all patient-driven innovations is diffused.[119] The drivers and barriers

to the diffusion and adoption of patient-driven innovations merit further investigation.

## Implications for policy and practice

As our findings show, patient-driven innovations have the potential to improve not only self-care processes, but also healthcare processes. However, although some of the publications indicated an interest among healthcare professionals in the opportunities offered by patient-driven innovations, only a few reported outcomes that reflected the use of patient-driven innovations in healthcare. This may be related to existing barriers (eg, liability, safety, ethics and policy) to the implementation of patient-driven innovations in healthcare.[58 120 121] Experts on health innovation and commercialisation processes, although not necessarily experts on patient-driven innovations, emphasise the importance of clear liability frameworks and governance of patient-driven innovations.[58] To address regulatory hurdles, a multidisciplinary approach involving multiple stakeholders (eg, end-users, healthcare professionals, manufacturers and regulators) has been recommended.[27] A concern is that decision-makers and policy-makers often underestimate the value of end-user innovations.[120 121] Thus, to successfully support regulatory approval and implementation of patient-driven innovations in healthcare may require an approach whereby healthcare professionals and researchers recognise and collaborate with patient innovators as independent innovation sources.[122] Patient design has been suggested as an approach to engage with patients as true partners in a collaborative process.[3] From previous research, there is evidence that cocreation with patients in the design and planning of care can benefit patient–professional relationships, improve health outcomes and also increase satisfaction among clinicians.[123] Further, partnering with patients in research has been associated with improved public health and health-related outcomes for patients.[124] Such cocreation or partnership processes are often driven by researchers or professionals. More efforts may be needed to support and study patient-driven cocreation processes.

## Unanswered questions and future research

We call for future studies to explore the transferability of our findings to innovations openly published in the social network space. A recent publication illustrates how machine learning methods can be applied to identify pioneering lead user innovations posted openly on the web that respond to unmet needs, so-called need-solution pairs.[125] Such an analysis would likely contribute to developing a more nuanced categorisation of unmet needs addressed by patient-driven innovations.

Further, although patient-driven innovations are gaining ground, our findings indicate that research on the outcomes of patient-driven innovations is still limited, which may prevent their adoption in other contexts, such as healthcare. Therefore, more quality evaluation studies may be needed to evaluate the effects of patient-driven innovations. This, in turn, may support the future adoption of effective patient-driven innovations in various contexts. As evaluation studies require collaboration with researchers and possibly other stakeholders, we could gain valuable insights by exploring the motives and experiences of patient innovators who have participated in such collaborations. Patient innovators were coauthors in some, but not all, publications included in our scoping review. Thus, it would be interesting to understand how patient innovators reason about actively engaging in scientific publishing. Future research should also explore how to build supportive structures for multidisciplinary collaborations that can contribute to quality assurance, evaluation and implementation of patient-driven innovations in healthcare.

## CONCLUSIONS

The patient-driven innovations reported in this study illustrate a diversity of innovative approaches that patients and informal caregivers have applied to address unmet needs related to their self-care, open sharing of information and knowledge with others, interactions with peers and collaborations with healthcare. Although positive impacts on self-care, as well as peer interactions, and healthcare collaboration were reported, outcomes of patient-driven innovations need further investigation. To support wider implementation, more research is also needed into regulatory approval processes, dissemination and governance of patient-driven innovations. This is important because as our findings illustrate, patients and informal caregivers are central stakeholders in driving healthcare development and research forward to meet the needs that truly matter to patients.

**Acknowledgements** The authors would like to thank the members in the research program 'Patients in the driver's seat!' for their valuable contributions in the analysis process.

**Contributors** All authors (MD, ML, MM, AB, HH, SR, MR and CW) contributed to the conceptualisation and design of the study and methodology; MD, ML and MM extracted and analysed the data with support from MR and SR; all authors contributed to interpreting the results; MR and HH drafted the introduction; MD drafted the remainder of the manuscript with supervision from CW; all authors revised or critically reviewed the manuscript and read and approved the final draft. HH was responsible for the overall content as the guarantor.

**Funding** This work was supported by the Swedish Research Council for Health, Working Life and Welfare/Forskningsrådet om Hälsa, Arbetsliv och Välfärd (FORTE) grant number 2018-01472.

**Competing interests** SR who coauthored this paper as a patient innovator and researcher is also an author of one of the publications that was included in the analysis.

**Patient and public involvement** Patients and/or the public were involved in the design, or conduct, or reporting, or dissemination plans of this research. Refer to the Methods section for further details.

**Patient consent for publication** Not applicable.

**Provenance and peer review** Not commissioned; externally peer reviewed.

**Data availability statement** All data relevant to the study are included in the article or uploaded as online supplemental information.

**ORCID iDs**
Marie Dahlberg http://orcid.org/0000-0001-9611-5256
Henna Hasson http://orcid.org/0000-0002-3827-6841
Maria Reinius http://orcid.org/0000-0003-0864-8701
Carolina Wannheden http://orcid.org/0000-0003-2122-1083

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
