## [Reviewer comments · BMJ Open]

ARTICLE DETAILS

TITLE (PROVISIONAL)	Objectives and outcomes of patient-driven innovations published in peer-reviewed journals: A qualitative analysis of publications included in a scoping review
AUTHORS	Dahlberg, Marie; Lek, Madelen; Malmqvist, Moa; Bylund, Ami; Hasson, Henna; Riggare, Sara; Reinius, Maria; Wannheden, Carolina

VERSION 1 – REVIEW

REVIEWER	Demonaco, Harold MIT Sloan Finance Group
REVIEW RETURNED	31-Jan-2023

GENERAL COMMENTS	The authors provide a manuscript detailing their efforts to better define patient driven innovations as published in the literature. They based their findings on an analysis of a previously obtained and published literature review. They further posit that unmet needs of patients may be categorized into one of three categories, (access to self-care support tools, open sharing of information and knowledge, and patient agency in self-care and healthcare decisions) and two categories of outcomes (impact on self-care, and impact on peer interaction and healthcare collaboration). To my knowledge this is the first such attempt at the development of a standardized nomenclature for patient innovation derived from the literature. In doing so, they potentially provide future research a valuable tool. However, because of their data source, their observations, and conclusions may not be generalizable. In their previously published article¹, they noted the difficulties and limitations associated with the identification of innovations by patients using the literature as a source. As noted by Canhao and colleagues², significant barriers exist, limiting patient innovators from diffusing their ideas and findings in the literature. Previous studies³ have demonstrated that patient innovation is commonplace with up to 7.9% of responders with a rare disease indicating development of a medically useful solutions for themselves. The literature suggests that innovators diffuse their finds via social networks.⁴ If this is the case, non-academics would not be likely to utilize the medical and organizational science literature to do so. Thus, a review of the published literature may not in fact represent the nature, substance, and market advantages of patient innovation. The paucity of findings in the literature as compared to the over 1200 innovations listed at the time of the study in Patient-Innovation.com speaks directly to the conundrum. The authors speak to this potential liability directly but do not do so adequately in my estimation. Additional justification for the use of the formal
--

	literature would be welcomed; specifically related to publication bias. The concept of free revealing is a central process in open-source software development. Innovators freely reveal the new and novel software code they have developed to fellow developers and to so-called free riders. Their reasons for doing so have been discussed elsewhere but inherent is the assumption that they in some way benefit directly. The benefits include a potential for improvements or identification of “bugs” by others. The personal costs then of free revealing is potentially offset by rewards. Social networks populated by other interested participants are presumably necessary for diffusion and improvements. This feedback loop is limited in the published literature. As anyone who has or has attempted to publish research results in the formal literature, the costs in time and energy can be significant. An examination of the references does not support the social network theory of diffusion. The citations are in large measure the work of a relatively small number of innovators and other interested parties diffusing information to their social network and on behalf of patients. Inherent in this process is a professional filtering process by clinicians, reviewers, and editors of formal journals. Given this filtering, it is entirely possible that the innovations that make it to publication represent a biased sample.⁵ The dominance of publications related to OPENAPS and Patients Like Me speak to this potential bias. Publication in the formal literature represents a formidable barrier to diffusion for both the lay public and experts in the field. Given this barrier, it is not surprising that there is a paucity of literature as compared to the self-reported prevalence of patient innovation reported in the literature based on direct representative sampling. Communities of practice (COP) possess an informal and fluid model of knowledge sharing centered on the testing, application and learning that comes from the experience of “solving recurring issues held in common⁵. To do so, they rely on multiple communication methods including Twitter, Reddit, Facebook, Autodesk Instructables and issue specific websites to name a few. A scan of health related projects described on Autodesk Instructables noted instructions for a host of healthcare related projects: automated pill dispensers, child therapy chair, assistive suspenders, numerous face shield designs, DIY portable ventilator, DIY electroencephalogram, custom orthopedic braces, and a feeding tube cover to name a few. These non-refereed sources of information may contain innovations that may not meet the formal refereed literature filter process. But may hold valuable information on nascent solutions. Other researchers have utilized web-based sources to identify user innovations utilizing several methodologies. A recent publication by von Hippel and Kaularz utilizing machine learning methods⁶ is an example. Although their research was conducted within the kite surfing community, the methods are presumably applicable to other domains. In summary, the authors have presented a research paper that utilizes appropriate methods, is well written and comprehensive. The conclusions drawn however are presumptive and not entirely supported by the data. The authors point out the significant limitations of their conclusions in an open and honest way. However, the generalizability of the conclusions is called into significant question based on the limited samples. Applying the proposed innovation categorizations in the social network space
--	--

	may provide for additional clarity and support the generalizability of these observations.  1. I suggest that the authors expand their comments on the significant limitations associated with the application of peer reviewed literature in the exploration of patient innovation. 2. Given the paucity of published articles related to patient innovation, what is the perceived value of the categories proposed? 3. In the “Unanswered Questions” section, the authors suggest, “... such studies require collaboration with researchers and other stakeholders...” Why is this collaboration necessary for the diffusion of patient innovation? While curated sites may provide a level of credibility, they represent a filtering system that may inhibit diffusion. 4. A review of multiple citations suggest that innovators were not authors. For example, several citations listed under “Diabetes” in Appendix 1 do not appear to have been authored by patient innovators (e.g. Murray, Oliver). Please justify their inclusion and the inclusion of other citations in additional categories not authored by the innovators themselves.  1. Reinius, M., Mazzocato, P., Riggare, S., Bylund, A., Jansson, H., Øvretveit, J., Savage, C., Wannheden, C. and Hasson, H., 2022. Patient-driven innovations reported in peer-reviewed journals: a scoping review. BMJ open, 12(1), p.e053735. 2. Canhao H, Zeinlovic L, Oliviera P. Revolutionizing healthcare by empowering patients to innovate. EMJ Innov 2017;1:31-34 3. von Hippel, E.A., Ogawa, S. and PJ de Jong, J., 2011. The age of the consumer-innovator. 4. Oliveira, P., Zejnilović, L. and Canhao, H., 2017. Challenges and opportunities in developing and sharing solutions by patients and caregivers: The story of a knowledge commons for the Patient Innovation project. governing medical knowledge commons, 301. 5. Lewis, D., 2022. Barriers to Citizen Science and Dissemination of Knowledge in Healthcare. Citizen Science: Theory and Practice, 7(1). 6. Wenger, E., 2000: Communities of practice and social learning systems. Organization, 7, 225–246 7. von Hippel, E. and Kaulartz, S., 2021. Next-generation consumer innovation search: Identifying early-stage need-solution pairs on the web. Research Policy, 50(8), p.104056.
--	---

REVIEWER	Meskó, Bertalan The Medical Futurist Institute
REVIEW RETURNED	07-Mar-2023

GENERAL COMMENTS	As the concept of patient design gains attention, it's really important to have this first qualitative analysis of published literature on patient-driven innovations. I applaud the authors for doing that. I only have a few questions/comments: It would be great if authors could comment in the paper on why certain medical conditions such as diabetes are more represented in patient-led innovations. Could another implication for policy and practice be embracing patient design?
---

	What could be methods for encouraging patient-led innovation on the patients' side?
--	---

REVIEWER	Nissling, Linnea University of Gothenburg, Department of Psychology
REVIEW RETURNED	15-Mar-2023

GENERAL COMMENTS	Thank you for getting the possibility to review the manuscript entitled "Objectives and outcomes of patient-driven innovations published in peer-reviewed journals: A qualitative analysis of publications included in a scoping review". The study addresses an important topic related to patient agency within health care and reports on a qualitative attempt to review which unmet needs that patient-driven health care innovations addresses and preliminary outcomes that have been reported on within the scientific literature. The study is overall well-done and nicely written. I have some comments and remarks mostly pertaining to the reporting on the methodology and results, as well as limitations that should be mentioned. Introduction Much of the introduction revolves around persons living with chronic conditions. Is your focus within this study on patient-driven innovations for chronic conditions only? In that case I think this should be made clearer, for example in the description of search criteria for the scoping review. Method I think the methods section could benefit from just a few summary sentences on how articles were selected for the scoping review that this study builds on, and some information on the included articles. As a reader you don't want to read previous articles for being able to assess this one. For example, a description of the search terms used, data bases searched, and number of independent interventions reported on in the included articles, country of origin etc. The authors clearly describe the procedure for analyzing the articles included for the study. However, I would wish for a description and reference for the method used for analysis – qualitative content analysis (f.e Graneheim & Lundman, 2004). Moreover, a motivation for using content analysis, as opposed to any other qualitative method, would be preferable. How did you reach conclusions about the themes based on the data? Were there any subthemes that were condensed or initial themes that were later abandoned? This could be elaborated on further in the manuscript. The authors describe that they collaboratively analyzed the articles and resolved any discrepancies in interpretation through discussion which is adequate. However, was there someone else in the research group/outside which wasn't involved in the coding
--

	process that could validate that the themes corresponded to the text? Hence, triangulating. I would want a more thorough description of the pre-understanding of the researchers, in terms of for example knowledge, experiences and perceptions, and how this could have influenced the interpretation of data. Is your analysis inductive or deductive? Do you analyze manifest or latent content? Decisions like this could be made clearer in the manuscript. Do you feel that your results and analysis are saturated and are the themes mutually exclusive? Results Is there enough data to arrive at the theme “impact on peer interaction and health care collaboration”? I feel that this is two different outcomes, one which has to do with social interactions with peers with similar experiences, and one which has to do with interaction with health care. Further, the outcomes for interaction with health care is based on only two studies. From “the subtheme’s title” it’s not clear how these patient-driven interventions affected peer relationships and communication with health care? Where there any objective changes that were reported in these studies? If so, this could be made clearer in the manuscript. Relatedly, one way to make this subtheme clearer could be to elaborate on the name of the theme a bit. As a suggestion you could make it clearer that it focuses on improved or deepened communication/collaboration; for example “Improved communication with peers and health care professionals” Discussion You discuss in the section for principal finding that the reported outcomes of patient-driven innovations should be interpreted with some caution as they are based on studies with varying designs (experimental, observational, and descriptive) and that you didn’t assess these studies for methodological quality. I agree with you, and would also highlight further limitations with these studies – for example: where these studies properly powered in terms of enough participants? Which outcome measures have they used to arrive at their conclusions – are they psychometrically sound? Moreover, given that only five out of 18 of these studies were RCTs, most of them probably can’t address causality or account for confounding variables. Therefore, I would address these limitations even more in the discussion section and use words like “preliminary effectivity” throughout the manuscript to make it clear that these interventions need to be assessed further. The latter can also be addressed in the section for future research, hence that not just more research is needed but also more quality research based on RCTs. Conclusion
--	---

	A minor reflection, in the first sentence in the conclusions, on page 17, you could add the “open sharing of information and knowledge with others” for the sentence to cover all of the identified unmet needs: The patient-driven innovations reported in this study illustrate a diversity of innovative approaches that patients and informal caregivers have applied to address unmet needs related to their self-care, open sharing of information and knowledge with others, interactions with peers, and collaborations with healthcare. Tables I would consider adding a figure with the respectively themes and subthemes to make the results more easily accessible. Possibly with some examples illustrating the themes. Abstract I would add something in the abstract as well about the uncertainty and limited research to date about the outcomes of patient-driven innovations that you report on.
--	---

VERSION 1 – AUTHOR RESPONSE

Reviewer 1 comments:

The authors provide a manuscript detailing their efforts to better define patient driven innovations as published in the literature. They based their findings on an analysis of a previously obtained and published literature review. They further posit that unmet needs of patients may be categorized into one of three categories, (access to self-care support tools, open sharing of information and knowledge, and patient agency in self-care and healthcare decisions) and two categories of outcomes (impact on self-care, and impact on peer interaction and healthcare collaboration). To my knowledge this is the first such attempt at the development of a standardized nomenclature for patient innovation derived from the literature. In doing so, they potentially provide future research a valuable tool. However, because of their data source, their observations, and conclusions may not be generalizable. In their previously published article¹, they noted the difficulties and limitations associated with the identification of innovations by patients using the literature as a source. As noted by Canhao and colleagues², significant barriers exist, limiting patient innovators from diffusing their ideas and findings in the literature. Previous studies³ have demonstrated that patient innovation is commonplace with up to 7.9% of responders with a rare disease indicating development of a medically useful solutions for themselves. The literature suggests that innovators diffuse their finds via social networks.⁴ If this is the case, non-academics would not be likely to utilize the medical and organizational science literature to do so.

Thus, a review of the published literature may not in fact represent the nature, substance, and market advantages of patient innovation. The paucity of findings in the literature as compared to the over 1200 innovations listed at the time of the study in Patient-Innovation.com speaks directly to the conundrum. The authors speak to this potential liability directly but do not do so adequately in my estimation. Additional justification for the use of the formal literature would be welcomed; specifically related to publication bias.

The concept of free revealing is a central process in open-source software development. Innovators freely reveal the new and novel software code they have developed to fellow developers and to so-called free riders. Their reasons for doing so have been discussed elsewhere but inherent is the

assumption that they in some way benefit directly. The benefits include a potential for improvements or identification of “bugs” by others. The personal costs then of free revealing is potentially offset by rewards. Social networks populated by other interested participants are presumably necessary for diffusion and improvements. This feedback loop is limited in the published literature. As anyone who has or has attempted to publish research results in the formal literature, the costs in time and energy can be significant.

An examination of the references does not support the social network theory of diffusion. The citations are in large measure the work of a relatively small number of innovators and other interested parties diffusing information to their social network and on behalf of patients. Inherent in this process is a professional filtering process by clinicians, reviewers, and editors of formal journals. Given this filtering, it is entirely possible that the innovations that make it to publication represent a biased sample.⁵ The dominance of publications related to OPENAPS and Patients Like Me speak to this potential bias.

Publication in the formal literature represents a formidable barrier to diffusion for both the lay public and experts in the field. Given this barrier, it is not surprising that there is a paucity of literature as compared to the self-reported prevalence of patient innovation reported in the literature based on direct representative sampling.

Communities of practice (COP) possess an informal and fluid model of knowledge sharing centered on the testing, application and learning that comes from the experience of “solving recurring issues held in common⁵. To do so, they rely on multiple communication methods including Twitter, Reddit, Facebook, Autodesk Instructables and issue specific websites to name a few. A scan of health related projects described on Autodesk Instructables noted instructions for a host of healthcare related projects: automated pill dispensers, child therapy chair, assistive suspenders, numerous face shield designs, DIY portable ventilator, DIY electroencephalogram, custom orthopedic braces, and a feeding tube cover to name a few.

These non-refereed sources of information may contain innovations that may not meet the formal refereed literature filter process. But may hold valuable information on nascent solutions. Other researchers have utilized web-based sources to identify user innovations utilizing several methodologies. A recent publication by von Hippel and Kaulartz utilizing machine learning methods⁶ is an example. Although their research was conducted within the kite surfing community, the methods are presumably applicable to other domains.

In summary, the authors have presented a research paper that utilizes appropriate methods, is well written and comprehensive. The conclusions drawn however are presumptive and not entirely supported by the data. The authors point out the significant limitations of their conclusions in an open and honest way. However, the generalizability of the conclusions is called into significant question based on the limited samples. Applying the proposed innovation categorizations in the social network space may provide for additional clarity and support the generalizability of these observations.

1. Reinius, M., Mazzocato, P., Riggare, S., Bylund, A., Jansson, H., Øvretveit, J., Savage, C., Wannheden, C. and Hasson, H., 2022. Patient-driven innovations reported in peer-reviewed journals: a scoping review. *BMJ open*, 12(1), p.e053735.

2. Canhao H, Zeinlovic L, Oliveira P. Revolutionizing healthcare by empowering patients to innovate. *EMJ Innov* 2017;1:31-34

3. von Hippel, E.A., Ogawa, S. and PJ de Jong, J., 2011. The age of the consumer-innovator.

4. Oliveira, P., Zejnilović, L. and Canhao, H., 2017. Challenges and opportunities in developing and sharing solutions by patients and caregivers: The story of a knowledge commons for the Patient Innovation project. *governing medical knowledge commons*, 301.

5. Lewis, D., 2022. Barriers to Citizen Science and Dissemination of Knowledge in Healthcare. *Citizen Science: Theory and Practice*, 7(1).

6. Wenger, E., 2000: Communities of practice and social learning systems. *Organization*, 7, 225–246

7. von Hippel, E. and Kaulartz, S., 2021. Next-generation consumer innovation search: Identifying early-stage need-solution pairs on the web. *Research Policy*, 50(8), p.104056.

1: I suggest that the authors expand their comments on the significant limitations associated with the application of peer reviewed literature in the exploration of patient innovation.

Author response: Thank you for your detailed and constructive comments and suggestions on how we could further clarify our limitations. With the support of your suggested references, we have expanded our comments on the limitations associated with our study sample that consists of peer-reviewed literature only. It would indeed be interesting to investigate the social network space to explore the generalizability of our findings and contribute to a more nuanced categorization of unmet needs addressed by patient-driven innovations. Our changes are found in the Strengths and limitations section (p. 10-11, lines 253-262) and the Unanswered questions and future research section (p. 12-13, lines 324-329).

2: Given the paucity of published articles related to patient innovation, what is the perceived value of the categories proposed?

Author response: Even though our study sample merely represents a fraction of the openly available patient-driven innovations, we believe that our categorization of unmet needs addressed by patient-driven innovations and reported outcomes may provide a valuable tool for future research, as you also suggested in your comment. The value is dependent on how we interpret the findings. We are aware that most patient-driven innovations are not published in peer-reviewed journals and therefore we cannot assume that our results are representative for the entire field. However, if this limitation is accepted and acknowledged, our findings can be used as a starting point to explore this phenomenon in more depth and also to explore the transferability of our findings in future studies. We have added comments about this in the Strengths and limitations section (p. 10, lines 250-251; 253-258, p. 11, lines 281-283) and the Unanswered questions and future research section (p. 12-13, lines 324-329).

3: In the “Unanswered Questions” section, the authors suggest, “.... such studies require collaboration with researchers and other stakeholders....” Why is this collaboration necessary for the diffusion of patient innovation? While curated sites may provide a level of credibility, they represent a filtering system that may inhibit diffusion.

Author response: We agree that collaboration with researchers and other stakeholders may not necessarily contribute to the diffusion of innovations to other end-users and could even be an inhibiting factor. As our findings show that research studies on outcomes of patient-driven innovations are limited, our intention was to suggest that collaboration with researchers may be necessary for increasing the number of evaluation studies. Evaluation studies that report effects of patient-driven innovations may in turn contribute to their adoption in various contexts (e.g., healthcare, where evidence is important for implementation). We have now rephrased these sentences to clarify our argument in the Unanswered questions for future research section (p. 13, lines 330-337).

4: A review of multiple citations suggests that innovators were not authors. For example, several citations listed under “Diabetes” in Appendix 1 do not appear to have been authored by patient innovators (e.g. Murray, Oliver). Please justify their inclusion and the inclusion of other citations in additional categories not authored by the innovators themselves.

Author response: Your observation is correct. Co-authorship was not an eligibility criterion in the scoping review. Our aim was to identify, describe, and analyze the nature of innovations that had been developed by patients and informal caregivers, but not necessarily published by them. However, we are currently performing a follow-up study that explores the drivers and experiences of patient authors. To avoid confusion, we have added a sentence about this in the Unanswered questions and future research section (p. 13, lines 337-339).

Reviewer 2 comments:

As the concept of patient design gains attention, it's really important to have this first qualitative analysis of published literature on patient-driven innovations. I applaud the authors for doing that.

I only have a few questions/comments:

1: It would be great if authors could comment in the paper on why certain medical conditions such as diabetes are more represented in patient-led innovations.

Author response: Unfortunately, we are unable to provide an explanation to why certain medical conditions are more represented in the sample of patient-driven innovations analyzed in our study. However, we have now expanded our comments about the limitations of our study, in particular related to study selection (p. 10, lines 256-262). We have also expanded on our Unanswered questions and future research section (p. 13, lines 337-339), in which we highlight the value of further exploring the motives and experiences of patient innovators who engage in collaborations with researchers and who decide to engage as patient authors.

2: Could another implication for policy and practice be embracing patient design?

Author response: Thank you for suggestion! We have added this under Implications for policy and practice (p. 12, lines 315-316)

3: What could be methods for encouraging patient-led innovation on the patients' side?

Author response: This is an interesting question that we are unfortunately also unable to answer. Exploring methods for encouraging patient-led innovation was beyond the scope of our study. However, we are currently performing a follow-up study in which we have interviewed patient innovators about their motives and experiences of publishing their innovations in scientific journals. Our upcoming study may contribute to more empirical data about how patient-led innovation can be encouraged and supported.

Reviewer 3 comments:

Thank you for getting the possibility to review the manuscript entitled "Objectives and outcomes of patient-driven innovations published in peer-reviewed journals: A qualitative analysis of publications included in a scoping review". The study addresses an important topic related to patient agency within health care and reports on a qualitative attempt to review which unmet needs that patient-driven health care innovations addresses and preliminary outcomes that have been reported on within the scientific literature. The study is overall well-done and nicely written. I have some comments and remarks mostly pertaining to the reporting on the methodology and results, as well as limitations that should be mentioned.

Introduction

1: Much of the introduction revolves around persons living with chronic conditions. Is your focus within this study on patient-driven innovations for chronic conditions only? In that case I think this should be made clearer, for example in the description of search criteria for the scoping review.

Author response: Thank you for this observation. We agree that it may need to be clarified that this study was not by design limited to patient-driven innovations for chronic conditions only. However, given the time and effort needed to develop and share health innovations, it is not surprising that most innovations concern the management of chronic conditions. This is supported by the studies included in the scoping review (which also was not limited to chronic conditions). We have added some clarifications in the Introduction (p. 4, lines 39-41; 66).

Method

2: I think the methods section could benefit from just a few summary sentences on how articles were selected for the scoping review that this study builds on, and some information on the included articles. As a reader you don't want to read previous articles for being able to assess this one. For example, a description of the search terms used, databases searched, and number of independent interventions reported on in the included articles, country of origin etc.

Author response: We certainly understand your concern. Meanwhile, word count limitations forced us to make difficult choices about what information to condense and elaborate on, respectively. Given that the selection criteria for our scoping review are available in our referenced citation, we decided to not replicate this information in the methods section of this study. By referring to the scoping study, we also avoid confusion about what was done in this qualitative study, compared to the scoping review. The study selection process was not a part of the current study. We hope that you accept our justification.

3: The authors clearly describe the procedure for analyzing the articles included for the study. However, I would wish for a description and reference for the method used for analysis – qualitative content analysis (f.e Graneheim & Lundman, 2004). Moreover, a motivation for using content analysis, as opposed to any other qualitative method, would be preferable.

Author response: We performed a qualitative content analysis, as described by Graneheim and Lundman (2004 & 2017), because this is a suitable method for summarizing large amounts of text, enabling us to stay close to the data. We have now clarified our motivation for using this method, including references (p. 5, lines 79-82).

4: How did you reach conclusions about the themes based on the data? Where there any subthemes that were condensed or initial themes that were later abandoned? This could be elaborated on further in the manuscript.

Author response: We chose to stay close to the manifest content of the data throughout the analysis process. The abstraction of meaning units into codes and categories was an iterative process that we continued until all agreed on the level of abstraction represented by the final categories. We hope that this has been sufficiently clarified by our changes in the Methods section (p. 6, lines 101-104).

5: The authors describe that they collaboratively analyzed the articles and resolved any discrepancies in interpretation through discussion which is adequate. However, was there someone else in the research group/outside which wasn't involved in the coding process that could validate that the themes corresponded to the text? Hence, triangulating.

Author response: Thank you for raising this question. The coding and analysis process was largely performed by three of the co-authors (MD, ML, MMC), as indicated in our methods description. The final categories were then presented to the other co-authors which led to iterative refinements. We have clarified this in the Methods section (p. 6, lines 101-104).

6: I would want a more thorough description of the pre-understanding of the researchers, in terms of for example knowledge, experiences and perceptions, and how this could have influenced the interpretation of data.

Author response: We appreciate your request for a more “thick description”, which we agree is important in qualitative research. At the same time, as we were many researchers involved, describing all researchers' pre-understanding in terms of knowledge, experience, and perceptions would not be feasible given the word limits. In line with COREQ guidelines, the researchers' credentials and professional backgrounds are disclosed. Additional details regarding the researchers' pre-understanding can be obtained through personal communication with the authors. We hope that you are satisfied with this explanation.

7: Is your analysis inductive or deductive? Do you analyze manifest or latent content? Decisions like this could be made clearer in the manuscript.

Author response: Thank you for highlighting this. We have now clarified that we performed an inductive manifest content analysis in the Methods section (p. 5, lines 79-82). Please see our response to your comment 3.

8: Do you feel that your results and analysis are saturated and are the themes mutually exclusive?

Author response: As the concept of saturation is closely tied to grounded theory and inconsistently applied, we chose to discuss the concept of information power (Malterud, Siersma & Guassora, 2016) instead. Irrespective of terminology, we thank you for raising a valid concern. We have therefore added additional reflections about the trustworthiness of our categories in relation to the information power of our sample. We have also emphasized that our categories are indicative and that assessing the transferability of our findings to a broader sample of patient-driven innovations is subject to future research. Strengths and limitations section (p. 11, lines 265-275; 281-283)

Although we aimed to create as distinct categories as possible, mutual exclusivity is not always possible, as also described by Graneheim and Lundman (2004). Thus, "Access to self-care support tools", for example, likely shares some characteristics with "Patient agency in self-care and healthcare decisions". These unmet needs are closely intertwined as the satisfaction of access to self-care support tools will also contribute to enhance patient agency, for example. Yet, we see value in these distinct categories as they describe what we identified as unmet needs that can be understood as separate, although intertwined.

Results

9: Is there enough data to arrive at the theme "impact on peer interaction and health care collaboration"? I feel that this is two different outcomes, one which has to do with social interactions with peers with similar experiences, and one which has to do with interaction with health care. Further, the outcomes for interaction with health care is based on only two studies. From "the subtheme's title" it's not clear how these patient-driven interventions affected peer relationships and communication with health care? Where there any objective changes that were reported in these studies? If so, this could be made clearer in the manuscript.

Author response: We agree that peer interaction and healthcare collaboration are two different outcomes although they both concern engaging in conversations and sharing of information, knowledge, and experiences with others. The data for the outcomes categories were relatively scarce, especially regarding healthcare collaborations, which was the reason for gathering the two outcomes in one theme. None of the studies reported any objective changes.

10: Relatedly, one way to make this subtheme clearer could be to elaborate on the name of the theme a bit. As a suggestion you could make it clearer that it focuses on improved or deepened communication/collaboration; for example "Improved communication with peers and health care professionals"

Author response: We appreciate your suggestion and agree that it would be favorable with more expressive category names. However, given the limitations with the outcomes categories that we have described, we were not able to draw firm conclusions about positive versus negative outcomes (e.g., whether communication actually improved or not). We could only conclude that impact on communication and collaboration was an outcome of interest.

Discussion

11: You discuss in the section for principal finding that the reported outcomes of patient-driven innovations should be interpreted with some caution as they are based on studies with varying designs (experimental, observational, and descriptive) and that you didn't assess these studies for methodological quality. I agree with you, and would also highlight further limitations with these studies – for example: where these studies properly powered in terms of enough participants? Which outcome measures have they used to arrive at their conclusions – are they psychometrically sound? Moreover, given that only five out of 18 of these studies were RCTs, most of them probably can't address causality or account for confounding variables. Therefore, I would address these limitations even more in the discussion section and use words like "preliminary effectivity" throughout the manuscript to make it clear that these interventions need to be assessed further. The latter can also

be addressed in the section for future research, hence that not just more research is needed but also more quality research based on RCTs.

Author response: Our study was based on a scoping review that aimed to explore the nature and proliferation of patient-driven innovations published in peer-reviewed journals. As scoping reviews address broader topics than systematic reviews that may include a relatively narrow range of quality assessed studies, it is common for scoping reviews to a) include various study designs, and b) not to assess included studies for methodological quality. We have now further justified this in the Strengths and limitations section (p. 11, lines 278-281). We believe that the concerns that you raise would be relevant for a future systematic review with more narrow research questions concerning the effects of patient-driven innovations. To highlight the importance of more quality evaluation studies, we have slightly rephrased our section about Unanswered questions and future research (p. 13, lines 330-334).

Conclusion

12: A minor reflection, in the first sentence in the conclusions, on page 17, you could add the “open sharing of information and knowledge with others” for the sentence to cover all of the identified unmet needs:

Author response: Thank you for your suggestion! We have included it accordingly in the Conclusions section (p. 13, line 346)

Tables

13: I would consider adding a figure with the respectively themes and subthemes to make the results more easily accessible. Possibly with some examples illustrating the themes.

Author response: Thank you! We agree that it is desirable to illustrate the findings in tables or figures. We have presented all the categories in Figures 1 and 2. Therefore, we believe that adding an additional figure illustrating our categorization would maybe be too much.

Abstract

14: I would add something in the abstract as well about the uncertainty and limited research to date about the outcomes of patient-driven innovations that you report on.

Author response: Thank you for pointing this out! We have clarified this in the Conclusion section of the abstract (p. 2, lines 21-23):